# Determinants of hyperemesis gravidarum among pregnant women attending health care service in public hospitals of Southern Ethiopia

Gedife Ashebir[1], Haymanot Nigussie[1], Mustefa Glagn[1], Kassaw Beyene[2]*, Asmare Getie[3]

1 School of Public Health, College of Medicine and Health Sciences, Arba Minch University, Arba Minch, Ethiopia, 2 Department of Midwifery, College of Medicine and Health Sciences, Arba Minch University, Arba Minch, Ethiopia, 3 School of Nursing, College of Medicine and Health Sciences, Arba Minch University, Arba Minch, Ethiopia

* kassaw.kb3@gmail.com

**Data Availability Statement:** All relevant data are within the manuscript and its Supporting Information files.

## Abstract

### Background

Hyperemesis gravidarum is severe nausea and excessive vomiting, starting between 4 and 6 gestational weeks, peak at between 8 and 12 weeks and usually improve and subside by 20 weeks of pregnancy. Identifying the determinants of hyperemesis gravidarum has a particular importance for early detection and intervention to reduce the health, psychosocial and economic impact. In Ethiopia there is low information on determinants of hyperemesis gravidarum.

### Methods

Institution based unmatched case-control study design was conducted from April 12- June 12, 2021. A structured face-to-face interviewer administered questionnaire and checklist for document review were used to collect the data from 360 study participants (120 cases and 240 controls). The data were collected by KoBocollect 1.3, and then exported to statistical package for social science version 25 for further analysis. Both bi-variable and multivariable logistic regression analysis were done to identify the determinants and a p-value < 0.05 with a 95% confidence level was used to declare statistical significance.

### Result

Being an urban dweller (AOR = 2.1, 95% CI: 1.01, 4.34), having polygamous husband (AOR = 2.92, 95% CI: 1.27, 6.68), having history asthma/ other respiratory tract infections (AOR = 3.56, 95% CI: 1.43, 8.82), saturated fat intake (AOR = 4.06 95% CI: 1.98, 8.3), no intake of ginger (AOR = 3.04 95% CI: 1.14, 8.09), and inadequate intake of vitamin B rich foods (2.2, 95% CI: 1.14–4.2) were the determinants of hyperemesis gravidarum.

**Funding:** The author(s) received no specific funding for this work.

**Competing interests:** The authors have declared that no competing interests exist.

**Abbreviations:** ANC, Ante Natal Care; AOR, Adjusted Odd Ratio; CI, Confidence Interval; HEG, Hyper Emesis Gravidarum; NVP, Nausea and Vomiting of Pregnancy.

## Conclusion

This study revealed that, urban residence, having polygamous husband, history of asthma/ other respiratory tract infections, intake of saturated fat, no intake of ginger, inadequate intake of vitamin B reach foods were found to be independent determinants of hyperemesis gravidarum. It is better if healthcare providers and government authorities exert continual effort to give health education and counselling service concerning to dietary practice and asthma attacks. It is advisable if pregnant women adhere to healthy diets and limit intake of saturated fats and also husband and nearby relatives give care and support for pregnant women.

## Introduction

Pregnancy is an important period in which physiological, psychological, and social changes are experienced, and this requires adaptation to these changes. The process of adaptation to the role of pregnancy and motherhood varies depending on the individual's memories, psychosocial economic, environmental conditions, wishes and physiological symptoms, complications resulting from pregnancy [1].

Nausea and vomiting of pregnancy (NVP) are very common in early pregnancy and it is considered as a part of normal physiology [2]. Up to 80% of all pregnant women experience some form of nausea and vomiting during their pregnancy [3].

According to The International Statistical Classification of Disease and Related Health Problems, hyperemesis gravidarum (HEG) is defined as 'persistent and excessive vomiting starting before the end of the 22nd week of gestation [4]. It is characterized by persistent vomiting and nausea at least three times per day, weight loss of more than 5% of pre-pregnancy body weight, ketonuria, electrolyte abnormalities and, dehydration, resulting in a poor quality of life and increased health care cost [5]. Hyperemesis gravidarum is the most common cause of hospitalization during the first half of pregnancy [6].

A cohort study conducted in Nova Scotia, Canada revealed that hyperthyroidism disorders, psychiatric illness, history of molar pregnancy, pre-existing diabetes mellitus, gastrointestinal disorder and, asthma were associated with increased risk of hyperemesis gravidarum [7].

A cross -sectional study conducted in Egypt showed gastrointestinal diseases, urinary tract infection and multiple pregnancies were the most common risk factors of hyperemesis gravidarum [8]. Un-unmatched case-control study conducted in Bale zone hospitals, indicates that being urban residence, being employed, being in the first trimester and second trimester period and having perceived stress illness were associated factors of HEG [9].

In Norway about 25% of women with HEG want to terminate the pregnancy and 75% of them prefer to stop getting pregnant again [10]. In Botswana, 2.4% pregnant women died because of hyperemesis gravidarum [11]. In different parts of Ethiopia, the magnitude of hyperemesis gravidarum 4.4% in Addis Ababa [12], 4.8% in Jima [13], and 8.2% in Arba Minch [14] were diagnosed.

HEG adversely affects physical activities and work performance [15], family and social relationships [15,16], psychological status [16,17], nutrition [18], and health of women, decreases their quality of life and makes adoption to pregnancy is difficult [19,20].

Further, it causes serious complications like pre-eclampsia placental abruption, coagulopathy, neuromuscular complications organ damage, and even death [21,22]. Women will become

dehydrated and no longer be able to provide the fetus with essential nutrients for growth which results in intrauterine pregnancy loss, growth restriction, intrauterine fetal death, pre-term delivery, low birth weight, low 5-minute Apgar score, and increase risks of neural tube defects [23,24].

Identifying the determinants of hyperemesis gravidarum have a particular importance for early detection and intervention to reduce the health, psychosocial and, economic impact on the women and families. Despite having many studies done elsewhere, in Ethiopia little is known about the predictors of hyperemesis gravidarum. Therefore, this study aimed to iden-tify determinants of hyperemesis gravidarum among pregnant women attending health care service in public hospitals of Gamo, Gofa, and South Omo Zones.

## Methods and materials

### Study setting and design

An institution-based case-control study design was conducted in public hospitals of southern Ethiopia, from April 12 to June 12, 2021. In the southern region, there are 15 zones, the study was conducted in three zones of 9 public hospitals of Gamo, Gofa, and south Omo. Arba Minch town is the administrative city of Gamo Zone, which is 505 km far from Addis Ababa. Sawla town is the administrative city of Gofa Zone, which is 464 km far from Addis Ababa, and Jinka town, which is the administrative city of South Omo Zone and 755 km far from Addis Ababa, the capital city of Ethiopia. Currently, in Gamo zone there are five Hospitals (Arba Minch Gen-eral Hospital, Chencha Hospital, Selamber hospital, Kemba and Gerese hospital). In Gofa zone there are two hospitals (Saula general hospital and Laha primary hospital). And South Omo zone has two public hospitals (Jinka general hospital and Gazer primary hospital).

**Source population.**   All pregnant women who attended health care services in public hos-pitals of Gamo, Gofa, and south Omo zones were the source population.

**Study population.**   *Cases*. Pregnant women in the antenatal period admitted with HEG in public hospitals of Gamo, Gofa, and south Omo zones during the study period were the cases.

*Controls*. Pregnant women attending antenatal care visit, and not diagnosed with HEG in public hospitals of Gamo, Gofa, and south Omo zones during the study period were the controls.

### Exclusion criteria

*Cases*. Pregnant women who are severely ill and unable to respond for the interview were excluded from the study.

*Controls*. Pregnant women, whoever treated for HEG in the current pregnancy were excluded from the study.

### Sample size determination and sampling techniques

The sample size was calculated by using EpiInfo version 7 menu StatCalc programs for four potential determinants which were significant in recent studies with the consideration of the following assumptions: confidence level 95%, power 80, and exposed to an unexposed ratio of 1:2. Which is taken from the previous study done in Bale zone, Ethiopia [9] on risk factors of hyperemesis gravidarum (by taking the factor being in the 2$^{nd}$ trimester when severe NVP or HEG starts) and the largest sample size was 327, and 10% of the total sample size was added to compensate non-response rate and the final sample size was 360 (120 cases and 240 controls).

To get the required number of cases and controls, proportional allocation was done to each hospital based on the number of women admitted for HEG. Cases were selected every other

mother (k = 2) until the sample size reached, for each case, two controls (k different for each hospital) were selected from pregnant women attending antenatal care visit by using systematic random sampling.

## Operational definition

Hyperemesis gravidarum. Hyperemesis gravidarum refers to intractable nausea vomiting during pregnancy that leads to weight loss and volume depletion, resulting in ketonuria [25].

History of hyperemesis gravidarum: when pregnant women claimed or documented as she was ever diagnosed or treated for HEG at least once in the previous pregnancy.

Cases. Defined as women in antenatal period clinically diagnosed by the physician as being hyperemesis gravidarum.

Controls. Defined as the women in the antenatal period that had not been diagnosed with hyperemesis gravidarum.

## Data collection tool and quality control

Data were collected from pregnant women attending health care service by using a structured face-to-face interviewer administered questionnaire and checklist for document review were used to collect the data from the study participants. Initially, it was prepared in English language then translated to the Amharic language and back to English to ensure consistency. The questionnaire contains socio-demographic characteristics, reproductive questions, medical factors, psychological factors and dietary factors. A 10 -item multiple-choice self-report psychological instrument was used for measuring the perception of stress. Each answer is scored 0 to 4, 0 –never, 1—almost never, 2 –sometimes, 3—fairly often, and 4—very often. Revers scoring was used for positive statement questions (4, 5, 6, and 7). It is scored by summing across all scale items. Scores ranging from 0–13 would be considered low stress, scores ranging from 14–26 would be considered moderate stress, and scores ranging from 27–40 would be considered high perceived stress [26].

Patient health questionnaire was used to assess depression; which comprises nine items that can be scored from 0 (not at all) to 3 (nearly every day). Scores ranging from 0–4 would be considered minimal depression, 5–9 mild depression, and 10–14 moderate depression, 15–19 moderately severe depression and 20–27 Severe depression [27]. Modified dietary history tool was used to assess dietary practice of pregnant women [28].

Two days training was given for nine data collectors and three supervisors. Then, pre-test was conducted on 5% [18] of the sample size. Cronbach's Alpha was calculated by using SPSS software version 25 to test internal consistency (reliability) of the item, and Cronbach's Alpha greater than 0.7 was considered as reliable and 0.949 for perceived stress illness and 0.839 for depression were obtained.

## Data processing and analysis

The data were collected by KoBocollect version 1.3 then exported to statistical package for social science (SPSS) version 25 for analysis. Frequency distribution table was used to for presentation of data. Bivariable analysis, crude odds ratio with 95% CI, was used to see the association between each independent variable and the outcome variable. Independent variables with p-value of $\leq 0.25$, biologically plausible and consistent in the previous study were included in the multivariable analysis to control confounding factors. Multicollinearity test was done before model fitness was assessed and VIF was less than 10. Hosmer and Lemeshow's goodness-of-fit test was checked, and it was found to be insignificant (p value = 0.821) which indicate the model was fitted. Finally, multivariable logistic regression analysis was done to assess

the determinants of hyperemesis gravidarum. Level of statistical significance was declared at p value < 0.05 with, 95% Confidence Interval.

## Ethical consideration

The study obtained ethical approval from Arba Minch University, College of Medicine and Health and Sciences, Institutional Research Ethics Review Board (IRB/1078/21). Based on the approval, an official letter was written by Arba Minch University Public Health Department to each Zonal Health Department. Explanation on the objective of the research was provided to public hospitals administrators. Similarly, the administrators of each public hospital wrote letter to the concerned unit. Then the respondents were informed about the purpose and procedure of the study, the importance of their participation, the benefits, and risks associated with the study, the right to withdraw at any time if they feel discomfort. After explaining the purpose of the study, written consent was obtained. To maintain the confidentiality of information gathered from the study participant, code numbers were used throughout the study. During each contact with study participants COVID19 transmission prevention measures were taken. For each data collector single reusable mask was provided.

## Results

### Socio-demographic characteristics of respondents

A total of 360 study participants (120 cases and 240 controls) were interviewed in the study. The mean age was 27.19 (SD±5.19) for cases and 26.53 (SD±5.33) for controls respectively. Nearly two third of women with hyperemesis gravidarum 89 (74.2%) and more than half of controls 147 (61.3%) were from urban areas. Ninety one percent of cases 109 (90.8%) and almost all controls 229 (95.4%) were married. Less than ten percent of cases 11 (9.2%) and thirteen percent of controls 30 (12.5%) have no formal education. More than two third of 81 (67.5%) cases and sixty three percent of controls 150 (62.5%) have no leisure time physical activity (**Table 1**).

### Obstetrics characteristics of respondents

Almost all cases 115 (95.8%) and nearly three percent 6(2.5) of controls were in the first trimester period. The mean gestational age was 8.18(SD±2.86) and 25.52(SD±6.27) for cases and controls respectively. More than half of the cases 64(53.3%) and more than one-third of controls 99(41.3%) hadn't the previous experience of pregnancy. The Majority of cases 103(85.8%) and controls 213 (88.8%) reported that their inter pregnancy interval was two years and above and the mean inter-pregnancy interval was 25.96(SD±5.67) months for cases and 28.34(SD±5.93) months for controls. Nearly one-third of cases 18(32.1%) and sixteen percent of controls 22 (15.6%) had history of hyperemesis gravidarum. Regarding to the current pregnancy about two-third of cases 81(67.5%) and more than three-fourth of controls 211(87.9%) reported that their pregnancy was planned. While large proportion of cases 112(93.3%) and controls 231 (96.3%) reported that their pregnancy was wanted.

About forty-five percent of women with hyperemesis gravidarum 25(44.6%) and about quarter 35(24.8%) of women without hyperemesis gravidarum had bad obstetric history (**Table 2**).

### Medical histories of respondents

Sixteen percent of women with hyper emesis gravidarum 19(15.8%) and four percent 9(3.8%) of pregnant women without hyperemesis gravidarum had previous history of diabetes mellitus.

**Table 1. The socio-economic characteristics of pregnant women attending health care service in public hospitals, southern Ethiopia, 2021.**

| Variables | | Cases(n = 120) | Controls(n = 240) |
|---|---|---|---|
| | | N% | N% |
| Age | ≤ 20 | 13(10.8) | 33(13.8) |
| | 21–25 | 34(28.3) | 82(34.2) |
| | 26–30 | 44(36.7) | 68(28.3) |
| | 31–35 | 22(18.3) | 43(17.9) |
| | 36 and above | 7(5.8) | 14(5.8) |
| Residence | Urban | 89(74.2) | 147(61.3) |
| | Rural | 31(25.8) | 93(38.8) |
| Ethnicity | Gamo | 62(51.7) | 132(55.0) |
| | Gofa | 27(22.5) | 52(21.7) |
| | Wolayita | 4(3.3) | 11(4.6) |
| | Konso | 4(3.3) | 5(2.1) |
| | Others[a] | 23(19.2) | 40(16.7) |
| Religion | Orthodox | 32(26.70 | 109(45.4) |
| | Protestant | 66(55.0) | 104(43.3) |
| | Muslim | 8(6.7) | 22(9.2) |
| | Others[b] | 14(11.7) | 5(2.1) |
| Marital status | Married | 109(90.8) | 229(95.4) |
| | Not living in marital union[c] | 11(9.2) | 11(4.6) |
| Having polygamous husband | Yes | 29(24.2) | 31(12.9) |
| | No | 91(75.8) | 209(87.1) |
| Educational status | No formal education | 11(9.2) | 30(12.5) |
| | Primary school | 43(35.8) | 56(23.3) |
| | Secondary school | 25(20.8) | 93(38.8) |
| | College and above | 41(34.2) | 61(25.4) |
| Occupation | Employee | 34(28.3) | 45(18.8) |
| | Merchant | 18(15.0) | 28(11.7) |
| | Farmer | 15(12.5) | 24(10.0) |
| | House wife | 42(35.0) | 113(47.1) |
| | Others[d] | 11(9.2) | 30(12.5) |
| leisure time physical activity | Three and more per week | 17(14.2) | 28(11.7) |
| | 1–2 per week | 12(10.0) | 21(8.8) |
| | 1–3 per month | 10(8.3) | 41(17.1) |
| | Never | 81(67.5) | 150(62.5) |

Others[a]: Aari, Male, Daasanach, Hamer, Banna, Amhara, and Tsamai Others[b]: Catholic and traditional religions, Not living in marital union[c]: Single, separated, divorced, widowed Others[d]: Students, and daily laborers, jobless.

About quarter of cases 30(25%) and five percent of controls 12(5.0%) had history of asthma or other respiratory tract infections. Ninety four percent of cases 112(93.3), and almost all controls 238(99.2%) had no history of hyperthyroid disorder (**Table 3**).

## Dietary characteristics of respondents

Nearly half of the cases 62 (51.7%) and three fourth of controls 174(72.5%) had a habit of eating snacks. More than-two third of cases 86(71.7%) and about half of the controls 125(52.1%) had a habit of eating spiced foods. More than three-fourth of cases 103(85.8%) and two third of controls 172(71.7%) had inadequate water intake. More than half of the cases 67(55.8%) and

**Table 2. Obstetric characteristics of pregnant women attending health care service in public hospitals, southern Ethiopia, 2021.**

| Variables | | Cases | Controls |
|---|---|---|---|
| | | N (%) | N (%) |
| Gestational age in month | First trimester | 115(95.8) | 6(2.5) |
| | Second trimester | 5(4.2) | 146(60.8) |
| | Third trimester | 0(0.0) | 88(36.7) |
| Gravidity | Primigravida | 64(53.3) | 99(41.3) |
| | Multigravida | 56(46.7) | 141(58.8) |
| Parity | Nulliparous | 2(3.6) | 16(11.3) |
| | Primiparous | 18(32.1) | 44(31.2) |
| | Multiparous | 36(64.3) | 81(57.4 |
| Number of alive children | Have no alive child | 3(5.4) | 4(2.8) |
| | 1–2 | 38(67.8) | 88(62.4) |
| | 3 and above | 15(26.8) | 49(34.8) |
| Planed pregnancy | Yes | 81(67.5) | 211(87.9) |
| | No | 39(32.5) | 29(12.1) |
| Wanted pregnancy | Yes | 112(93.3) | 231(96.3) |
| | No | 8(6.7) | 9(3.8) |
| Supported pregnancy | Yes | 104(86.7) | 229(95.4) |
| | No | 16(13.3) | 11(4.6) |
| bad obstetric history | Yes | 25(44.6) | 35(24.8) |
| | No | 31(55.4) | 106(75.2) |
| Inter-pregnancy interval in a month | Less than 24 | 17(14.2) | 27(11.3) |
| | 24 and above | 103(85.8) | 213(88.8) |
| History Multiple pregnancies | Yes | 9(16.1) | 12(8.5) |
| | No | 47(83.9) | 129(91.5) |
| History of molar pregnancy | Yes | 4(7.1) | 2(1.4) |
| | No | 52(92.9) | 139(98.6) |
| History of HEG | Yes | 18(32.1) | 22(15.6) |
| | No | 38(67.9) | 119(84.4) |
| History gestational hypertension | Yes | 17(30.4) | 15(10.6) |
| | No | 39(69.6) | 126(89.4) |

fifteen percent of controls 37(15.4%) weren't iodized salt users during their pregnancy. Nearly half of the cases 57 (47.5%) and fifteen percent of controls 35(14.6%) hadn't ginger intake during their pregnancy. More than three-fourth of women with HEG 99(82.5%) and nearly half of

**Table 3. Psychological characteristics of pregnant women attending health care service in public hospitals, southern Ethiopia, 2021.**

| Variables | | Cases(n = 120) | Controls(n = 240) |
|---|---|---|---|
| | | N% | N% |
| Fear of reoccurrence of bad obstetric history | Yes | 19(76.0) | 20(57.1) |
| | No | 6(24.0) | 15(42.9) |
| Perceived stress | Low stress | 15(12.5) | 51(21.3) |
| | Moderate stress | 62(51.7) | 130(54.2) |
| | Sever stress | 43(35.8) | 59(24.6) |
| Depression | Mild depression | 15(12.5) | 76(31.7) |
| | Moderate depression | 45(37.5) | 131(54.6) |
| | Moderately severe depression | 53(44.2) | 26(10.8) |
| | Severe depression | 7(5.8) | 7(2.9) |

controls 113(47.15%) had a history of saturated fat intake. Nearly three-fourth of cases 87 (72.5%) and half of the controls 121(50.4%) hadn't adequate intake of vitamin B reach foods during their pregnancy.

## Determinants of hyperemesis gravidarum

Bivariate logistic regression was done between independent variables and HEG to identify candidate variables for multivariable logistic regression. residence, educational status, marital status, occupation, having polygamous husband, gravidity, history of Asthma/other respiratory tract infection, planned pregnancy, supported pregnancy, saturated fat intake, intake of vitamin B reached foods, ginger intake, water intake and eating seasoned foods had association on bivariate analysis. Those variables with a P value of $\leq 0.25$ in the bivariate analysis was entered to multivariable logistic regression model.

The result showed that, the odds of developing hyperemesis gravidarum was 2.1 (AOR = 2.1, 95% CI: 1.01, 4.34) times higher among urban dwellers as compared to rural dwellers. Mothers having polygamous husband were 2.92(AOR = 2.92, 95% CI: 1.27, 6.68) times at higher odds of developing HEG as compared to their counter parts. The odds of developing HEG was 3.56 (AOR = 3.56, 95% CI: 1.43, 8.82) times higher among Mothers having history asthma as compared to having no history of asthma or other RTI. The odds of developing hyperemesis gravidarum among pregnant women who had history of saturated fat intake was 4.06 (AOR = 4.06 95% CI: 1.98, 8.3) times higher as compared to had no history saturated fat intake. Pregnant women who had no intake of ginger were 3.04 (AOR = 3.04 95% CI: 1.14, 8.09) times at higher odds of developing HEG as compared to their counter parts. The odds of developing hyperemesis gravidarum among was 2.2(2.2, 95% CI: 1.14–4.2) times higher among pregnant women who had no adequate intake of vitamin B reach foods as compared to those who had adequate intake of vitamin B reach foods (**Table 4**).

**Table 4. Multivariable logistics regressions results for determinants of HEG among pregnant women attending health care service in public hospitals, southern Ethiopia, 2021.**

| Variables | Case N (%) | Control N (%) | COR (95%CI) | AOR (95%CI) |
|---|---|---|---|---|
| Residence | | | | |
| Urban | 89(74.2) | 147(61.3) | 1.82(1.12–2.95) | 2.1(1.01–4.37) |
| Rural | 31(25.8) | 93(38.8) | 1 | 1 |
| Having polygamous husband | | | | |
| Yes | 29(24.2) | 31(12.9) | 2.15(1.22–3.77) | 2.92(1.27–1.68) |
| No | 91(75.8) | 209(87.1) | 1 | 1 |
| History of Asthma/other RTI | | | | |
| Yes | 30(25.0) | 12(5.0) | 6.33(3.1–12.92) | 3.56(1.43–8.82) |
| No | 90(75.0) | 228(95.0) | 1 | 1 |
| Educational status of the women | | | | |
| No formal education | 11(9.2) | 30(12.5) | 0.55(0.25–1.20) | 1.33(0.45–3.94) |
| Primary education | 43(35.8) | 56(23.3) | 1.14(0.65–2.00) | 0.46(0.13–1.61) |
| Secondary education | 25(20.8) | 93(38.8) | 0.40(0.221–.724) | 0.92(0.23–3.71) |
| College and above | 41(34.2) | 61(25.4) | 1 | 1 |
| Occupation of the women | | | | |
| Employed | 34(28.3) | 45(18.8) | 1 | 1 |

*(Continued)*

**Table 4.** (Continued)

| Variables | Case N (%) | Control N (%) | COR (95%CI) | AOR (95%CI) |
|-----------|-----------|---------------|-------------|-------------|
| Merchant | 18(15.0) | 28(11.7) | 0.85(0.41–1.79) | 1.52(0.44–5.24) |
| Farmer | 15(12.5) | 24(10.0) | 0.83 (0.38–1.81) | 0.87(0.21–3.6) |
| House wife | 42(35.0) | 113(47.1) | 0.49(0.28-.87) | 0.76(0.27–2.13) |
| Others® | 11(9.2) | 30(12.5) | 0.49 (0.21–1.10) | 0.94(0.28–3.2) |
| Marital status of the women | | | | |
| Married | 109(90.8) | 229(95.4) | 1 | 1 |
| Not living in marital union | 11(9.2) | 11(4.6) | 2.10 (0.88–5.00) | 1.79(0.56–5.78) |
| Gravidity | | | | |
| Primigravida | 64(53.3) | 99(41.3) | 1.63 (1.05–2.53) | 1.55(0.86–2.79) |
| Multigravida | 56(46.7) | 141(58.8) | 1 | 1 |
| Planed pregnancy | | | | |
| Yes | 81(67.5) | 211(87.9) | 1 | 1 |
| No | 39(32.5) | 29(12.1) | 3.50(2.03–6.04) | 1.75(0.8–3.83) |
| Supported pregnancy | | | | |
| Yes | 104(86.7) | 229(95.4) | 1 | 1 |
| No | 16(13.3) | 11(4.6) | 3.2(1.44–7.14) | 1.26(.37–4.22) |
| Eating seasoned foods | | | | |
| Yes | 62(51.7) | 174(72.5) | 2.33(1.45–3.73) | 0.82(0.41–1.65) |
| No | 58(48.3) | 66(27.5) | 1 | 1 |
| Saturated fat intake | | | | |
| Yes | 99(82.5) | 113(47.1) | 5.3(3.10–9.05) | 4.1(1.98–8.3) |
| No | 21(17.5) | 127(52.9) | 1 | 1 |
| Intake of ginger | | | | |
| Not use | 57(47.5) | 35(14.6) | 6.51(3.11–13.64) | 3.04(1.14–8.1) |
| Sometimes | 50(41.7) | 153(63.7) | 1.31(.66–2.6) | 0.95(0.38–2.41) |
| Always | 13(10.8) | 52(21.7) | 1 | 1 |
| Water intake per day | | | | |
| Adequate | 17(14.2 | 68(28.3) | 1 | 1 |
| Inadequate | 103(85.8) | 172(71.7) | 2.35(1.31–4.21) | 1.26(0.62–2.58) |
| Vitamin B reached food intake | | | | |
| Adequate | 33(27.5) | 119(49.6) | 1 | 1 |
| Inadequate | 87(72.5) | 121(50.4) | 2.59(1.61–4.17) | 2.19(1.14–4.19) |

## Discussion

The finding of this study revealed that urban residence, having polygamous husband, having history of asthma/other RTI, Intake of saturated fat, no intake of ginger, and inadequate intake of vitamin B reach food were the determinants of hyperemesis gravidarum.

The result indicates that, the odds of developing hyperemesis gravidarum were two times higher among urban dwellers as compared to rural dwellers. This result supported by previous studies conducted in bale Zone hospitals [9], on the other hand, this finding is in contrast with the findings of studies conducted in Turkey [29], which concludes that there are no statistically significant differences between cases and controls concerning the residence. The possible explanation might be due to the difference housing conditions, environmental sanitation, sewerage system, and ventilation between turkey and Ethiopia [30].

Mothers having polygamous husband were three times at higher odds of developing HEG as compared to their counter parts. But this result is, on the contrary, to the study conducted in the Batman State Hospital, which showed that, there are no statistical differences between cases and controls in terms of having a polygamous husband [31]. This might be due to a difference in moral acceptability of polygamous marriage by the community and also care and support given to the women by the polygamous husband. Polygamous marriage can reduce self-esteem, marital satisfaction, and leads to marital conflict, somatization, depression, and anxiety; in turn those psychological problems can induce nausea and vomiting during pregnancy.

The odds of developing HEG were four times higher among Mothers having history asthma as compared to having no history of asthma or other respiratory tract infection. This result is in agreement with the study conducted in Nova Scotia, Canada [7], which concludes that pregnant women with past medical history of asthma and other respiratory disorders were found to be more liable to hospitalization due to HEG. The possible reason for this might be those women who had asthma can have a severe cough, during uncontrollable cough, repeated chest muscle contraction and relaxation puts pressure and disturb the stomach and finally it can trigger nausea and vomiting [32].

The odds of developing hyperemesis gravidarum among pregnant women who had a history of saturated fat intake were four times higher as compared to had no a history saturated fat intake. This study is in line with the study conducted in Boston, which found that Mothers having a history saturated fat intake were three times at higher odds of developing HEG compared to having no a history of saturated fat intake [33]. The possible explanation for this finding goes to the effect of saturated fat intake on circulating estrogen level. Saturated fat has been shown to increase circulating levels of estrogen. If the liver is clogged with too much saturated fat, it will have a hard time to breaking down estrogen in the body and estrogen will recirculate leading to the estrogen excess [34]. Estrogen contributes to HEG by stimulating the production of nitric oxide via nitrogen oxidase synthase, which in turn relaxes smooth muscle, slowing gastric intestinal transit time and gastric emptying [35]. A diet high in saturated fat also triggers inflammatory bowel diseases, consequently women more likely to experiencing nausea and vomiting [36].

Pregnant women who had no intake of ginger were three times at higher odds of developing HEG compared to have intake of ginger always. This result supports the report of a meta- analysis and literature review [37,38]. Which concludes ginger is an effective preventive and non-pharmacological option for the treatment of hyperemesis gravidarum. This may be related to the blocking effect of ginger on receptor cells. Ginger can antagonize activation of m3 muscarinic receptor and serotonin(5-HT3) receptors, thereby inhibiting afferent inputs to the central nervous system that are stimulated by specific neurotransmitters, released from the gastrointestinal tract [39,40]. Ginger also important for the digestion process, it works increasing Agni or 'digestive fire', which further helps to better break down and assimilation of food. Apart from this, ginger is also known to stimulate saliva, bile and gastric enzymes that aid digestion and help speed the movement of food from the stomach to the small intestine [41].

The odds of developing hyperemesis gravidarum was two times higher among pregnant women who had no adequate intake of vitamin B reach foods as compared to those who had adequate intakes of vitamin B rich foods. This result supports the findings of the research conducted in Sweden [42] which indicates that 28 percent of pregnant women who had intake of vitamins in early pregnancy were at low risk of developing HEG. This finding also in line with the study conducted in Norway [43] which showed that adherence to a vitamin b rich diet associated with a lower risk of developing hyperemesis. Vitamin B helps to prevent nausea and vomiting by prevent infections, promote healthy brain function and regulate and promote

good appetite, and facilitates/ease the digestion process by breakdown carbohydrate, fats and alcohol [44].

## Conclusion

This study found that urban residence, having polygamous husband, history of asthma/other respiratory tract infection, intake of saturated fat, no intake of ginger, and inadequate intake of vitamin b rich foods were important determinants of hyperemesis gravidarum. Healthcare providers should exert continual effort to give health education and counselling service concerning to dietary practice and asthma attacks. It is better if pregnant women adhere to healthy diets and limit intake of saturated fats and it is crucial to create awareness about the health hazards of saturated fat intake on health of pregnant women through multiple communication channels.

## Supporting information

**S1 File. Data collection tool.**
(DOCX)

**S2 File. The dataset used for this study.**
(SAV)

## Acknowledgments

The authors thank all the study participants and data collectors.

## Author Contributions

**Conceptualization:** Gedife Ashebir, Haymanot Nigussie, Mustefa Glagn, Kassaw Beyene, Asmare Getie.

**Data curation:** Gedife Ashebir, Haymanot Nigussie, Mustefa Glagn, Kassaw Beyene, Asmare Getie.

**Formal analysis:** Gedife Ashebir, Haymanot Nigussie, Mustefa Glagn, Kassaw Beyene, Asmare Getie.

**Funding acquisition:** Haymanot Nigussie, Asmare Getie.

**Investigation:** Gedife Ashebir, Haymanot Nigussie, Kassaw Beyene.

**Methodology:** Gedife Ashebir, Mustefa Glagn.

**Project administration:** Gedife Ashebir, Mustefa Glagn.

**Resources:** Mustefa Glagn.

**Software:** Gedife Ashebir, Haymanot Nigussie, Mustefa Glagn, Asmare Getie.

**Supervision:** Gedife Ashebir, Haymanot Nigussie, Mustefa Glagn, Kassaw Beyene, Asmare Getie.

**Validation:** Gedife Ashebir, Haymanot Nigussie, Mustefa Glagn, Kassaw Beyene, Asmare Getie.

**Visualization:** Gedife Ashebir, Haymanot Nigussie, Mustefa Glagn, Kassaw Beyene, Asmare Getie.

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
