## [Decision Letter · Decision Letter 0]

21 Dec 2021

PONE-D-21-31620Determinants of hyperemesis gravidarum among pregnant women attending health care service in public hospitals of Gamo, Gofa, and South Omo zones, Southern EthiopiaPLOS ONE

Dear Dr. Kassaw Beyene Getahun,

Thank you for submitting your manuscript to PLOS ONE. After careful consideration, we feel that it has merit but does not fully meet PLOS ONE’s publication criteria as it currently stands. Therefore, we invite you to submit a revised version of the manuscript that addresses the points raised during the review process.

ACADEMIC EDITOR:Dear authors on your scholarly work; you have brought an important study problem with good findings that have public health importance in the area of practice. However, the manuscript has multiple language usage flaws including punctuations, wordings, spelling and mainly grammar errors. These problems are found throughout the manuscript. Moreover, there are several methodological gaps. Therefore, please make repeated proof-reading and thorough copyediting before considering the manuscript for publication. This would help increase the readability of the manuscript if published.

We look forward to receiving your revised manuscript.

Kind regards,

Wubet Alebachew Bayih, M.Sc.

Academic Editor

PLOS ONE

Journal Requirements:

Reviewers' comments:

Reviewer's Responses to Questions

**Comments to the Author**

1. Is the manuscript technically sound, and do the data support the conclusions?

Reviewer #1: Partly

2. Has the statistical analysis been performed appropriately and rigorously? 

Reviewer #1: Yes

3. Have the authors made all data underlying the findings in their manuscript fully available?

Reviewer #1: Yes

4. Is the manuscript presented in an intelligible fashion and written in standard English?

Reviewer #1: No

5. Review Comments to the Author

Reviewer #1: Find the attached document. Regarding on the manuscript, I attached all the review findings in a document way, as per the author will try to react with feedbacks. All the comments should be taken as critical to be amended

6. PLOS authors have the option to publish the peer review history of their article (what does this mean?). If published, this will include your full peer review and any attached files.

Reviewer #1: No

---

## [Author Response · Author response to Decision Letter 0]

18 Jan 2022

We would like to say thank you very much for both the reviewer and academic editor for your critical review and constructive comments to result in scientifically sounded research finding.

---

## [Decision Letter · Decision Letter 1]

10 Feb 2022

PONE-D-21-31620R1Determinants of hyperemesis gravidarum among pregnant women attending health care service in public hospitals, Southern EthiopiaPLOS ONE

Dear Dr. Kassaw Beyene,

Thank you for submitting your manuscript to PLOS ONE. After careful consideration, we feel that it has merit but does not fully meet PLOS ONE’s publication criteria as it currently stands. Therefore, we invite you to submit a revised version of the manuscript that addresses the points raised during the review process.

ACADEMIC EDITOR: 

Please submit your revised manuscript by March 24/2022.  If you will need more time than this to complete your revisions, please reply to this message or contact the journal office at plosone@plos.org. Please include the following items when submitting your revised manuscript:A rebuttal letter that responds to each point raised by the academic editor and reviewer(s). You should upload this letter as a separate file labeled 'Response to Reviewers'.A marked-up copy of your manuscript that highlights changes made to the original version. You should upload this as a separate file labeled 'Revised Manuscript with Track Changes'.An unmarked version of your revised paper without tracked changes. You should upload this as a separate file labeled 'Manuscript'.

We look forward to receiving your revised manuscript.

Kind regards,

Wubet Alebachew Bayih, M.Sc.

Academic Editor

PLOS ONE

Reviewers' comments:

Reviewer's Responses to Questions

**Comments to the Author**

1. If the authors have adequately addressed your comments raised in a previous round of review and you feel that this manuscript is now acceptable for publication, you may indicate that here to bypass the “Comments to the Author” section, enter your conflict of interest statement in the “Confidential to Editor” section, and submit your "Accept" recommendation.

Reviewer #1: All comments have been addressed

Reviewer #2: (No Response)

2. Is the manuscript technically sound, and do the data support the conclusions?

Reviewer #1: Partly

Reviewer #2: Yes

3. Has the statistical analysis been performed appropriately and rigorously? 

Reviewer #1: Yes

Reviewer #2: Yes

4. Have the authors made all data underlying the findings in their manuscript fully available?

Reviewer #1: No

Reviewer #2: No

5. Is the manuscript presented in an intelligible fashion and written in standard English?

Reviewer #1: Yes

Reviewer #2: No

6. Review Comments to the Author

Reviewer #1: Thank you very much for reviewing this paper.

I am pleased to recommend publication.

It needs proofreading before publication

Reviewer #2: Title of the research: Determinants of hyperemesis gravidarum among pregnant women attending health care service in public hospitals of Gamo, Gofa, and South Omo zones, Southern Ethiopia

Comments to authors

Hyperemesis gravidarum is a pregnancy problem with more of physiological origin and it has been well studied area. But, still, it can be a research agenda of today with strong arguments. I see that the authors’ arguments in this regard need further attention. The manuscript is not well crafted and it has several editorial problems that demand authors attention to push forward for publication.

Abstract

1. The background section of the abstract requires reconsideration. For instance, how studying the determinants of HEG could enhance early detection of HEG? In fact, it may be beneficial for early intervention to prevent occurrence of HEG, and further health and other damages once it happens.

2. What is the implication of having polygamous husband and HEG? How could you justify this relationship?

3. Line 24… delete coma: “psychosocial and, economic impact.”

4. Line 28…add spacing: “to collect the data from360 study participants”

5. Line 29…check spelling: “Kobcollect 1.3”. Even in this document I see spelling inconsistency for the same software stated here in the abstract and in the method sections… “Kobkollect”. The correct may be “KoBoCollect…”

6. Line 35…unnecessary capitalization: “Saturated fat intake”

7. The authors should rephrase the recommendation. The role that the government authorities have to play and even the word itself is vaguely stated. Do they really have to involve or expected to involve in health education and counselling?

Introduction

8. The introduction is not well synthesized and lacks coherence. For example, the first paragraph is long and taken from a single reference. Another example, 4th and 5th paragraphs are about risk factors of HEG while 6th and 7th paragraphs are about consequences of HEG. In the 8th paragraphs the authors come again to discuss about risk factors of HEG.

9. The authors didn’t show the magnitude of HEG in Ethiopia in general and in study area in particular. Despite knowing the consequential outcomes of HEG, understand the magnitude of HEG is very helpful for readers to have clear image in study area context.

10. Line 64 through 72 are about risk factors of HEG. But the evidences are not synthesized. The authors presented the contents of each sources cited to make up an independent sentence.

11. The arguments given in the last paragraph are not convincing. Even the factors are presented very grossly…sociodemographic factors, medical factors, obstetric factors, etc. Please try to address them in detail and in narrower scope. The sample size issue is also not clear. Was that not scientifically sound? How that happen? The authors also raise the argument that study areas were not included in the previous study…what is special for the study area? Anything that interests you to study in the stated sites/area?

12. Line 88… vague description: “A similar study also concludes that demographic and obstetric…” The authors say similar study but with different citation, in this case, what is the linking word similar is referring to? In terms of what? Was that in terms of study design or …?

13. Line 90…grammatical concern: “On the contrary, other studies on the contrary conclude that demographic…”

14. The issue of early detection must be reconsidered here also.

Methods

15. Too many subheadings, the authors can use relevant subheading to organize the contents under fewer subheadings.

16. The control definition and exclusion criteria. Did the selection of the controls and cases are done regardless of the gestational age or trimester?

17. The author should give citation for the article used to calculate the sample size and also include a description about where the referred study was done. Factors for NVP Vs HEG: were the factors you considered in sample size calculation are for NVP or HEG? If you used factors for NVP, how much they are relevant to the outcome of interest?

18. Line 104…coma misplaced. It is corrected as “Arba Minch Town, which is the Capital of the Gamo Zone, is 505…”. Similar correction is needed in line 107: “Sawla Town, which is the Capital of Gofa Zone, is 464”. Check line 110 for the same problem.

19. Sampling Techniques: Regarding case and control selection, the authors already stated that they used a systematic sampling after proportional allocation. But the K used at each health facility was unique, how this happened is not clear? Proportional allocation Vs Different K?

20. The author should provide very clear detail about each variable measurement especially those variables having psychometric properties such as stress, depression, etc; variable requiring clinical skill and dietary intake related assessments. The data collection methods used are also required to be clear. Nothing was stated about the data collection methods used in the method section except the crude insight given in the abstract section.

21. The authors should give attention for the Cronbach’s alpha value given in the data quality management section. What is the implication of a Cronbach’s alpha value above 0.90? The reliability given for perceived stress is 0.949.

22. The author shall provide detail information about data analysis. For example, candidate selection variable strategies for multivariable logistic regression were not explained. The issue of multicollinearity checks before fitting multivariable logistic regression model need to be addressed.

23. There are editorial problems…unnecessary bolding, inconsistent font size, grammatical issues, check how hyphen used, etc.

Results

24. The authors didn’t show how they come up with these determinants. They should clearly indicate the candidate variables for the multivariable logistic model. Then, they can go for result interpretations as they did now. Overall, how the multivariable logistic model was fitted is overlooked both in the method and in the results sections.

25. In Table 4, the authors presented p-values, for what these p-values stands for is my question? Is that for COR or AOR? What is the relevance of having p-value in the presence of confidence intervals for odds ratios? The table design is not even attractive for reader.

26. Line 222…error: “A total of 360 study participants (120 cases and 2240 controls)”

27. Line 224…spacing: 89(74.2 %)

28. There are many editorial problems… inconsistent font size, grammatical issues, etc.

Discussion

29. The discussion requires a major revision. It is better if you begin with brief summary of the main findings in the first paragraph. The results are brought here in the discussion without significant paraphrasing. Besides, the possible justifications or the implications of results given for the observed association are not convincing or non-relevant in some cases. Even some of the comparisons are vague to understand. For example, the comparison given for the positive association between urban residence and HEG are studies which conclude about the association between HEG and demographic factors…which demographic factors? Was that about residence? At the same time the justifications given are also not relevant. Make things very clear as much as possible.

Conclusion

30. The authors should revise the recommendation.

7. PLOS authors have the option to publish the peer review history of their article (what does this mean?). If published, this will include your full peer review and any attached files.

Reviewer #1: No

Reviewer #2: **Yes: **Dabere Nigatu

---

## [Author Response · Author response to Decision Letter 1]

1 Mar 2022

Thank you very much for your critical and constructive comments. you appreciated every things from minor to major error, and it is important to make the article scientific and readable. finally we are happy if you consider the paper for publication.

---

## [Decision Letter · Decision Letter 2]

14 Mar 2022

Determinants of hyperemesis gravidarum among pregnant women attending health care service in public hospitals of Southern Ethiopia.

PONE-D-21-31620R2

Dear Dr. Beyene,

We’re pleased to inform you that your manuscript has been judged scientifically suitable for publication and will be formally accepted for publication once it meets all outstanding technical requirements.

Kind regards,

Wubet Alebachew Bayih, M.Sc.

Academic Editor

PLOS ONE

Additional Editor Comments (optional):

Reviewers' comments:

Reviewer's Responses to Questions

**Comments to the Author**

1. If the authors have adequately addressed your comments raised in a previous round of review and you feel that this manuscript is now acceptable for publication, you may indicate that here to bypass the “Comments to the Author” section, enter your conflict of interest statement in the “Confidential to Editor” section, and submit your "Accept" recommendation.

Reviewer #2: All comments have been addressed

2. Is the manuscript technically sound, and do the data support the conclusions?

Reviewer #2: Yes

3. Has the statistical analysis been performed appropriately and rigorously? 

Reviewer #2: Yes

4. Have the authors made all data underlying the findings in their manuscript fully available?

Reviewer #2: Yes

5. Is the manuscript presented in an intelligible fashion and written in standard English?

Reviewer #2: Yes

6. Review Comments to the Author

Reviewer #2: (No Response)

7. PLOS authors have the option to publish the peer review history of their article (what does this mean?). If published, this will include your full peer review and any attached files.

Reviewer #2: **Yes: **Dabere Nigatu

---

## [Editor Report · Acceptance letter]

31 Mar 2022

PONE-D-21-31620R2 

Determinants of hyperemesis gravidarum among pregnant women attending health care service in public hospitals of Southern Ethiopia. 

Dear Dr. Beyene:

I'm pleased to inform you that your manuscript has been deemed suitable for publication in PLOS ONE. Congratulations! Your manuscript is now with our production department. 

Kind regards, 

on behalf of

Dr. Wubet Alebachew Bayih 

Academic Editor

PLOS ONE